# Hydrogel Surface-Modified Polyurethane Copolymer Film with Water Permeation Resistance and Biocompatibility for Implantable Biomedical Devices

**DOI:** 10.3390/mi12040447

**Published:** 2021-04-16

**Authors:** Hey In Jeong, Dae Hyeok An, Jun Woo Lim, Taehoon Oh, Hojin Lee, Sung-Min Park, Jae Hyun Jeong, Jae Woo Chung

**Affiliations:** 1Department of Organic Materials and Fiber Engineering, Soongsil University, 369 Sangdo-ro, Dongjak-gu, Seoul 06978, Korea; vjov@naver.com (H.I.J.); shavros@naver.com (D.H.A.); oth2140@naver.com (T.O.); 2Department of Chemical Engineering, Soongsil University, 369 Sangdo-ro, Dongjak-gu, Seoul 06978, Korea; ljw9424@soongsil.ac.kr (J.W.L.); nfejjh@ssu.ac.kr (J.H.J.); 3School of Electronic Engineering, Soongsil University, 369 Sangdo-ro, Dongjak-gu, Seoul 06978, Korea; hojinl@ssu.ac.kr; 4Department of Creative IT Engineering, Pohang University of Science and Technology (POSTECH), Pohang 37673, Korea; sungminpark@postech.ac.kr

**Keywords:** copolymer, hydrogel, implantable medical device, package, polyurethane (PU), poly(vinyl pyrrolidone) (PVP)

## Abstract

To use implantable biomedical devices such as electrocardiograms and neurostimulators in the human body, it is necessary to package them with biocompatible materials that protect the internal electronic circuits from the body’s internal electrolytes and moisture without causing foreign body reactions. Herein, we describe a hydrogel surface-modified polyurethane copolymer film with concurrent water permeation resistance and biocompatibility properties for application to an implantable biomedical device. To achieve this, hydrophobic polyurethane copolymers comprising hydrogenated poly(ethylene-co-butylene) (HPEB) and aliphatic poly(carbonate) (PC) were synthesized and their hydrophobicity degree and mechanical properties were adjusted by controlling the copolymer composition ratio. When 10 wt% PC was introduced, the polyurethane copolymer exhibited hydrophobicity and water permeation resistance similar to those of HPEB; however, with improved mechanical properties. Subsequently, a hydrophilic poly(vinyl pyrrolidone) (PVP) hydrogel layer was formed on the surface of the polyurethane copolymer film by Fenton reaction using an initiator and crosslinking agent and the effect of the initiator and crosslinking agent immobilization time, PVP concentration and crosslinking agent concentration on the hydrogel properties were investigated. Finally, MTT assay showed that the hydrogel surface-modified polyurethane copolymer film displays excellent biocompatibility.

## 1. Introduction

Interest in biomedical devices incorporating state-of-the-art engineering technology and life-science technology is gradually increasing with the trend of ultra-aging and living standards improvement in modern society [1,2]. In particular, implantable biomedical devices such as neurostimulators, electrocardiograms and blood glucose sensors that are inserted into the human body have attracted significant attention for more comfortable and precise human healthcare [3,4]. Such devices are already positioned as key biomedical devices to extend life and achieve the better quality of life that modern society demands. To insert such a biomedical implant device into the human body, the device must operate stably even in the moisture and electrolytes present in the human body and must not trigger any foreign body reactions in the body [5,6,7]. To this end, the exterior material used to wrap (protect) the biomedical device is very important. Currently, metals such as titanium are generally used as exterior materials for implantable biomedical devices; however, such metals pose a number of limitations including high cost, difficult processing and no flexibility, which may be inconvenient when used inside the human body [8,9]. Thus, research on packaging implantable biomedical devices using polymers at low cost and good flexibility with easy manufacturing processes is being actively conducted [10,11].

Among the many known polymers, polyurethane is one of the polymers used in various biomedical materials, owing to its excellent elasticity, elongation and easy processability [10,11,12,13]. In particular, because its physical properties can be adjusted by controlling the type and content of the soft and hard segments in the polyurethane chain, polyurethane can be used to create biomedical products with tailor-fit properties suitable for applications such as artificial blood vessels and organs [14,15]. However, it is difficult for polyurethane to prevent water permeation, owing to the polar functional urethane groups repetitively present in its chain; hence, it is difficult to protect internal electronic circuits or batteries from biological moisture [16,17]. To solve this problem, it is necessary to effectively prevent moisture permeability and diffusion by using a highly hydrophobic polyurethane. Unfortunately, when such a material is placed in the body, hydrophobic proteins can be deposited on the polyurethane surface, thereby forming a biofilm. This may lead to blood clot formation and inflammation by foreign body reaction, which may be fatal [18,19,20,21]. Thus, to develop a polyurethane-based implantable biomedical device exterior material, the development of a material that suppresses water permeation and simultaneously improves biocompatibility is highly desirable. In addition, mechanical durability must be considered, as it is necessary to protect the internal circuit of the biomedical device from external forces.

The most common methods used to impart biocompatibility to polyurethane are grafting of a hydrophilic polymer such as poly(ethylene glycol), polyglycerol and poly(vinyl pyrrolidone) (PVP) and formation of a hydrogel on the polyurethane surface [22,23,24,25]. Among them, PVP is one of the most widely used polymers in the manufacture of non-biodegradable hydrogels and is attracting much attention in biofield research and industry because of its useful biological properties, such as biocompatibility, non-antigenicity and non-toxicity [26,27,28]. Moreover, besides imparting biocompatibility, when the hydrogel is formed on the polyurethane surface using PVP, antibiotics can be contained inside the PVP hydrogel, thereby suppressing the initial inflammation that may occur after implantation of the device in the human body [29,30]. However, in this case, an appropriate hydrogelation method that does not damage the device during the surface modification of polyurethane is necessary, because the surface of the final product including the internal electronic device must be modified.

In this study, a polyurethane copolymer with excellent mechanical properties and high hydrophobicity was synthesized using hydrogenated poly(ethylene-co-butylene) (HPEB) and polycarbonate (PC). Subsequently, a PVP hydrogel was formed on the surface of the polyurethane copolymer through Fenton reaction, resulting in a hydrogel surface-modified polyurethane copolymer film that displays both water permeation resistance and biocompatibility. The PVP hydrogel formed by Fenton reaction is able to maintain a stable hydrogel because of its high moisture content and can be very easily applied to medical devices because it can be covalently fixed to the polyurethane surface within a short time [5]. Thus, the hydrogel surface-modified polyurethane copolymer film shows great potential for application as a soft packaging material for an implantable biomedical device capable of stably working for a long time in the body.

## 2. Materials and Methods

### 2.1. Materials

HPEB diol (MW = 3100 g mol^−1^) and aliphatic PC diol (MW = 3000 g mol^−1^) were purchased from Cray Valley and UBE Korea, respectively. Methylene diisocyanate (MDI), 1,4-butanediol (BD), dibutyltin dilaurate (DBTDL), PVP (MW = 360 kDa), ascorbic acid (AA), iron(II) chloride (FeCl_2_), ethylene glycol dimethacrylate (EGDMA), cumene hydroperoxide (CHP) and sodium dodecyl sulfate (SDS) were purchased from Sigma Aldrich and used without purification. The HPEB and PC diols were used after dehydration for 8 h in an 80 °C vacuum oven, while the other materials were used as received without further purification.

### 2.2. Methods

#### 2.2.1. Synthesis of the Polyurethane Copolymers

HPEB and the PC diols were dissolved in chloroform at 60 °C for 2 h under nitrogen flow and reflux condensation conditions. MDI and a catalytic amount of DBTDL were added to the solution and stirred vigorously for 3 h to form HPEB and the PC prepolymers. The BD chain extender was next added to the prepolymer solution and stirred for 4 h, after which the reaction was terminated with methanol. The reaction product was precipitated in methanol, filtered and vacuum dried at 60 °C overnight, to form an HPEB-PC polyurethane copolymer. The diol-to-MDI-to-BD feed molar ratio was 1:2:1 and the copolymer composition was tuned by controlling the HPEB-to-PC diol feed ratio. The detailed feed molar ratios of the reactants used in the polymerization of the polyurethane copolymers are listed in Table 1 and the synthetic scheme of the polyurethane copolymer is shown in Figure 1. The synthesized polyurethane copolymers were melt-pressed under 145 °C and 7000 psi conditions for 10 min to obtain a film with a thickness of approximately 0.7 mm.

#### 2.2.2. Surface Hydrogelation of the Polyurethane Copolymer Film

The polyurethane copolymer film was immersed in a hexane solution containing EGDMA and CHP at 25 °C. The film was removed from the hexane solution and then immersed in a distilled water solution containing PVP, FeCl_2_ (0.1%, *w/v*) and AA (0.1%, *w/v*). This formed a hydrophilic PVP hydrogel layer on the surface of the hydrophobic polyurethane copolymer film through radical polymerization by Fenton reaction. The resulting hydrogel surface-modified polyurethane copolymer film was washed with distilled water for 24 h, after removing any unreacted substances from the distilled water solution containing SDS (0.1%, *w/v*) and subsequently dried at room temperature for 24 h.

#### 2.2.3. Characterization

To analyze the chemical structures of the polyurethane copolymers, ^1^H NMR spectra were recorded at 400 MHz in CDCl_3_ using a Bruker Avance 400 spectrometer. The functional groups of the polyurethane copolymers were characterized by Fourier-transform infrared (FT-IR) spectroscopy using a Bruker VERTEX 70 spectrometer with a spectral resolution of 2 cm^−1^ over the range 4500–600 cm^−1^, using attenuated total reflectance (ATR). Gel permeation chromatography (GPC) was carried out at 35 °C in HPLC-grade chloroform (flow rate = 1.0 mL min^−1^) on a Young-In Chromass YL9100 series chromatograph equipped with a refractive index detector and three Young-In Styragel columns. The number-average molecular weight (*M*_n_) and molecular weight distribution (*M*_w_/*M*_n_) were calculated by calibration with polystyrene standards. Differential scanning calorimetry (DSC; Discovery DSC 25, TA Instruments) was performed under nitrogen flow. After preheating to 150 °C, the samples were cooled and subsequently reheated from −90 to 150 °C at a heating rate of 10 °C min^−1^. Dynamic mechanical analysis (DMA; DMA 8000, Perkin-Elmer) was conducted in tension mode on 23 (L) × 6 (W) × 0.7 (D) mm^3^ rectangular strips at a frequency of 1 Hz, at temperatures between −100 and 80 °C at a heating rate of 3 °C min^−1^, with an oscillatory strain of 5 μm, under nitrogen atmosphere. Tensile testing was carried out using a universal testing machine (UTM; DR UTM 100, Dr-Tech) with a 10 kgf load cell at a strain rate of 100 mm min^−1^ and room temperature, according to the ASTM D 638-V 100 specifications; the specimens were tested at least five times to ensure reproducibility. To evaluate the water permeation resistance, the water contact angle, water uptake and water vapor transmission rate (WVTR) were next measured. The water contact angle was evaluated using a DSA 100 instrument (Krüss, Hamburg, Germany). A 4 μL portion of distilled water was dropped five times at different positions on the film and the average value was calculated. To measure the water uptake, samples with the dimensions 60 (L) × 10 (W) × 0.7 (D) mm^3^ were immersed in deionized water at 37 °C and weighed regularly at different time intervals until constant weight gain. The water uptake was then calculated according to Equation (1) [31]:(1)$$W\left(\%\right)=\frac{W_{t}-W_{d}}{W_{d}}\times 100$$
where *W**_d_* is the weight of the completely dried sample (vacuum oven for 24 h) and *W**_t_* is the weight of the sample immersed in water at time t. The WVTR values were measured using a Permatran-W 3/33 MA instrument (Mocon) at 38 ± 2 °C and 100% relative humidity for 24 h, according to ASTM F1249. The dimensions of the sample for the WVTR measurements were 100 (L) × 100 (W) × 0.5 (D) mm^3^. X-ray photoelectron spectroscopy (XPS) measurements were performed using a Kratos Axis photoelectron spectrometer (Kratos Analytical, Manchester, UK) at a background pressure of approximately 1.0 × 10^−9^ Torr, using Mg Kα X-rays as the excitation source (1253.6 eV). All binding energies were calibrated relative to the C 1s peak at 284.5 eV. To evaluate the hydrogel amount (HG) and equilibrium swelling ratio (ESR) of the PVP hydrogel formed on the surface of the polyurethane copolymer film, the polyurethane copolymer films were first prepared in the dimensions 40 (L) × 40 (W) × 0.7 (D) mm^3^. Subsequently, the surface of the polyurethane copolymer film was modified with PVP hydrogel and immersed in distilled water for 24 h until the equilibrium swelling state was reached. The swollen gels were then wiped superficially with filter paper to remove any surface-bound water and immediately weighted. The HG and ESR values were calculated according to Equations (2) and (3) [32,33], respectively:(2)$$\mathrm{HG}\;\left({g\;m}^{-2}\right)=\frac{W_{d}-W_{0}}{S}$$
(3)$$\mathrm{ESR}\;\left(\%\right)=\frac{W_{eq}-W_{d}}{W_{d}-W_{0}}\times 100$$
where *W*_0_ is the weight of the polyurethane film sample measured after complete drying in a vacuum oven at room temperature for 24 h, *W*_d_ is the weight of the completely dried hydrogel-modified samples (vacuum oven for 24 h) and *W**_eq_* is the weight of the swollen hydrogel-modified sample after 24 h. Cell viability was qualitatively evaluated in optical density (OD) using an MTT assay [3-(4,5-dimethyl-2-thiazolyl)-2,5-diphenyl-2H-tetrazolium bromide; Sigma Aldrich] by incubating the 96 well plates in a 10% solution of M106 at 37 °C and 5% CO_2_ for 4 h. Human dermal fibroblast (HDFn) cells were proliferated in growth medium comprising Medium 106 (M106, Gibco, Waltham, MA, USA), 10% low serum growth supplement (LSGS, Gibco) and 1% penicillin-streptomycin (P/S, 100×, Biowest, Paris, France) and exposed to the hydrogel surface-modified polyurethane copolymer film for 3, 9 and 24 h. After incubating for 4 h, lysis buffer solution [50% (*w/v*) of *N*,*N*-dimethylformamide (DMF) in distilled water and 10% (*w/v*) SDS] was added to each well to terminate the reaction. The 96 well plates were incubated at room temperature for 2 h to allow formazan to diffuse into the medium. The absorbance was measured at 570 nm using a microplate reader (Thermo Fisher Scientific, Waltham, MA, USA).

## 3. Results and Discussion

Figure 2a shows the FT-IR spectrum of the synthesized polyurethane copolymer. Urethane amine (−NH) functional groups were observed at 3300 cm^−1^ and 1520 cm^−1^, while the peak at 2250 cm^−1^, characteristic of the isocyanate (−N=C=O) functional group of the MDI end-functionalized prepolymers, disappeared. 

In addition, peaks corresponding to the carbonyl (C=O) and ether (C–O–C) functional groups in the urethane bond were observed at 1700 and 1230 cm^−1^ respectively. [34] These results show that the urethane bond was formed through the reaction between the isocyanate at the end of the prepolymer and the hydroxyl group (–OH) of the chain extender, BD, indicating successful polyurethane polymerization. In particular, the IR spectrum intensity corresponding to the carbonyl (1740 cm^−1^) and ether (1240 cm^−1^) functional groups of the carbonate increased with increasing PC-to-HPEB feed ratio. [34] This indicates that the PC component in the main chain of the hydrophobic polyurethane copolymer increased with increasing input amount of PC. On the other hand, GPC measurements of the synthesized polyurethane copolymers showed that all the copolymers had similar molecular weights and molecular weight distributions, except for PC100, which is a polyurethane that solely comprises PC (Figure 2b and Table 2). Thus, we predicted that the physical properties of the polyurethane copolymer would not be dominated by the molecular weight but by the composition of the polyurethane copolymer. 

^1^H NMR measurements were conducted to qualitatively determine the HPEB-to-PC composition ratio in the polyurethane copolymer. As shown in Figure 2c, NMR peaks corresponding to MDI, PC and HPEB are well observed, ref. [35] confirming that the polyurethane copolymer was well prepared. In particular, the composition of the polyurethane copolymers was obtained by comparing the integral ratio between the hydrogen peak at the methyl site [−CH_3_; position #4 in Figure 2c] on the HPEB side chain and the hydrogen peak at the methylene site [–CH_2_–; position #6 in Figure 2c] of the main PC chain. The results revealed that a random copolymer was synthesized, in which the actual composition ratio of the copolymer was near-similar to the HPEB-to-PC feed ratio. This signifies that the desired composition and physical properties of the copolymer can be easily obtained by adjusting the HPEB-to-PC feed ratio.

DSC, DMA and tensile tests were next performed to investigate the thermal and mechanical properties of the polyurethane copolymer. As shown in Figure 3a,b, HPEB100 and PC100, polyurethanes that solely comprise HPEB and PC, respectively, presented single glass temperatures (*T*_g_) of −48.8 and −23.5 °C, respectively.

Considering that the *T*_g_ was −55 °C for HPEG-diol and −40 °C for PC-diol in the oligomer state before polymerization, the increase in *T*_g_ of HPEB100 and PC100 reveals the extension of the main chain and an increase in the molecular weight, thereby indicating successful polymerization of the polyurethane copolymer. Interestingly, when the PC content in the polyurethane copolymer was less than 20 wt%, the *T*_g_ of PC did not appear and only the *T*_g_ of HPEB was observed. In contrast, when the PC content was more than 20 wt%, the *T*_g_ values of both HPEB and PC appeared simultaneously. Moreover, as the PC content in the polyurethane copolymer increased, these two *T*_g_ values became more pronounced, indicating that phase separation occurred between HPEB and PC [36]. Notably, HPEB and PC displayed partial miscibility only below a certain content, because the non-polar HPEB and polar PC have poor miscibility from an enthalpy point of view. To observe this further, the tan δ value of the polyurethane copolymer was measured by DMA. As can be seen in Figure 3c, when the PC content in the polyurethane copolymer was less than 20 wt%, α-transition was observed at a temperature similar to that of HPEB100; however, α-transition by PC was not observed. On the other hand, for PC contents equal or greater than 20 wt%, the peak corresponding to the α-transition of HPEB appeared at −35 °C, while a gentle and round α-transition peak appeared simultaneously between −30 °C and 20 °C. We supposed that the α-transition between −30 °C and 20 °C was the α-transition caused by PC. Thus, we considered that the phase separation between the HPEB and PC chains occurred and the α-transitions corresponding to each component appeared simultaneously, as shown by the DSC data.

Tensile testing was next conducted using a UTM, to investigate the effect of the polyurethane copolymer composition on the mechanical properties. As shown in Figure 3d, the polyurethane comprising sole HPEB (HPEB100) showed a tensile strength of approximately 6.6 MPa and an elongation of approximately 530%. On the other hand, for the polyurethane comprising sole PC (PC100), a tensile strength of 32.8 MPa and elongation of 963% were observed, indicating excellent mechanical properties. The superior mechanical properties of PC100, which has a relatively smaller molecular weight, over those of HPEB100, were attributed to the polar carbonyl groups in the PC chain, which allowed higher intermolecular interaction with the PU chain than is observed with the non-polar HPEB. As a result, in the case of the polyurethane copolymers, in which HPEB and PC coexist simultaneously, the mechanical properties improved with increasing PC content, which can be explained by the polar interactions formed by the aforementioned PC. However, when the PC content exceeded 20 wt%, the tensile strength and elongation decreased again. In particular, for HPEB50C50, with a PC content of 50 wt%, the tensile strength and elongation were significantly lower than those of HPEB100. This was attributed to the intensification of the phase separation between HPEB and PC, even if the amount of mechanically excellent PC increased in the polyurethane copolymer [37]. Owing to the phase interface that lacked mutual attraction, these phase interfaces act as the weakest point, resulting in easier breakage of the polyurethane copolymer. 

To investigate the effect of the HPEB-to-PC composition ratio in the polyurethane copolymer on the resistance to water, water contact angle, moisture content and water permeation tests were performed. As shown in Figure 4a, HPEB100 and PC100 showed contact angles of approximately 104.3° and 84.6°, respectively. This indicates that the hydrophobicity of HPEB100 is relatively higher than that of PC100. Both HPEB100 and PC100 are polyurethanes comprising numerous urethane polar functional groups in the main chain. However, HPEB100 is made from non-polar olefin HPEB oligomers, whereas PC100 is made from polar PC oligomers with a carbonyl polarity every 5–6 carbon atoms. These differences resulted in HPEB100 having a lower polarity and higher hydrophobicity than PC100, limiting its ability to hydrogen bond with water. On the other hand, PC100 formed hydrogen bonds with water more easily than HPEB100. As a result, as the amount of PC in the polyurethane copolymer increased, the water contact angle gradually decreased. This can be explained by the gradual increase in the ratio of PC chains capable of hydrogen bonding with water, as mentioned above. Indeed, as shown in Figure 4b, the water absorption significantly increased with increasing PC amount in the copolymer. However, the water absorption of HPEB95C05 and HPEB90C10 with PC contents of 5 and 10 wt%, respectively, had almost the same value as that of HPEB100. This tendency was also observed in the water permeation tests. As shown in Figure 4d, the water permeabilities of PEB95C05 and HPEB90C10 were near-identical to that of HPEB100. This was attributed to the relatively low amount of PC present in these two copolymers. We supposed that the polar and hydrophilic characteristics of PC were somewhat shielded by the hydrophobic HPEB because PC was present in small amounts. In this case, PC was comparatively miscible with HPEB. 

As previously mentioned, to use a polymer as an implantable device packaging material, the material must display high hydrophobicity to prevent moisture from the body from penetrating into the device. Moreover, the packaging material must have adequate mechanical durability to reliably protect the electronic circuit. Considering the DSC, DAM, tensile test, contact angle, moisture content and water permeation test results, we concluded that the polyurethane copolymer that concurrently satisfied the water permeation resistance and mechanical durability requirements was HPEB90C10 in the polyurethane copolymers synthesized in this study. Indeed, this copolymer presented a water resistance near-similar to that of HPEB100. Moreover, there was no phase separation between HPEB and PC and thus, the mechanical durability was enhanced. However, even if the hydrophobic HPEB90C10 is suitable for preventing water permeation and has adequate mechanical durability, when a hydrophobic material is implanted in the body, it may trigger a foreign body reaction, causing inflammation and adversely affecting the human body. Hence, our next aim was to impart biocompatibility to HPEB90C10 by forming a hydrophilic hydrogel layer on the surface of the hydrophobic HPEB90C10 film. Figure 5a shows a schematic image of the manufacturing process for the hydrogel surface-modified polyurethane copolymer film. To modify the HPEB90C10 film surface with hydrogel, EGDMA and CHP, as the crosslinking agent and initiator, respectively, were first anchored onto the surface of the HPEB90C10 film using hexane treatment. Subsequently, a PVP hydrogel crosslinked with EGDMA was formed on the film surface via Fenton reaction. Figure 5b,c show the FT-IR and XPS spectra recorded before and after hydrogel formation on the surface of the HPEB90C10 film. As shown in the FT-IR spectrum in Figure 5b, PVP C=O and C−N peaks, which were not observed before the modification, were observed at 1648 cm^−1^ and 1286 cm^−1^, respectively, the EGDMA C−O peak was also observed at 1274 cm^−1^. [38] The XPS spectrum in Figure 5c shows that the nitrogen peak was not clearly observed before modification, but its intensity greatly increased after the PVP hydrogel was formed on the surface of the HPEB90C10 film. This was presumably due to the increase in the number of amine atoms following the introduction of PVP on the HPEB90C10 film surface. Furthermore, the oxygen intensity in the XPS spectrum also increased significantly due to the increase in the ether and carbonyl functional oxygens of the crosslinking agent (EGDMA) used to form the hydrogel. These results showed that the PVP hydrogel layer crosslinked with EGDMA was successfully formed on the surface of the hydrophobic HPEB90C10 copolymer, resulting in a hydrogel surface-modified polyurethane copolymer film.

The effect of the hexane treatment time and PVP and EGDMA concentrations on the characteristics of the hydrogel formed on the surface of the HPEB90C10 copolymer film were investigated. Figure 6a shows the amount of hydrogel and swelling ratio as a function of hexane treatment time, whereby the PVP and EGDMA concentrations were fixed at 5 wt%. As mentioned above, the crosslinking agent and cumene initiator (EGDMA and CHP, respectively) were slightly infiltrated into the surface and immobilized on the surface of the HPEB90C10 film during the hexane treatment. As shown in Figure 6a, the hexane treatment time did not significantly affect the amount of hydrogel and the swelling ratio of the hydrogel surface-modified HPEB90C10 film. However, the mechanical properties of the polyurethane copolymer gradually decreased with increasing a hexane treatment time (Appendix A). Thus, we considered 2.5 min as the optimal treatment time. 

Figure 6b shows the results of the hydrogel amount and swelling ratio with increasing PVP concentration. The hexane treatment time and concentration of the EGDMA crosslinking agent were fixed at 2.5 min and 5 wt%, respectively. When the PVP concentration was too low (2.5 wt%), the surface hydrogel was not uniformly formed and the hydrogel content and swelling ratio were very low. When the PVP concentration was increased to 5 wt%, the amount of hydrogel and degree of swelling increased significantly, indicating that the hydrogel was well formed on the surface of the HPEB90C10 film. However, when the PVP concentration was higher than 5 wt%, the hydrogel content increased but the swelling ratio reached equilibrium and then slightly decreased at 10 wt%. This was attributed to the high-concentration PVP solution, which reduced the diffusion of the iron catalyst and enhanced the hydrogel film density. As a result, when the PVP concentration exceeded 5 wt%, the degree of swelling did not increase any further, even though the degree of hydrogel formation still increased. In particular, when the PVP concentration was 10 wt%, the hydrogel film was thick and cracked after drying. Accordingly, we next fixed the hexane treatment time at 2.5 min and the PVP concentration at 5 wt% and measured the amount of hydrogel and swelling ratio with increasing EGDMA crosslinker concentration (Figure 6c). Similar to our previous observations, the amount of hydrogel and swelling ratio both increased up to a concentration of 5 wt% EGDMA, indicating that the hydrogel was well formed up to 5 wt% EGDMA. However, with a further increase in the concentration, both the degree to which the amount of hydrogel increased and the swelling ratio were reduced. This can be explained by the increase in the degree of crosslinking when EGDMA concentration was greater than 5 wt%, which hindered the swelling of the PVP hydrogel. When measuring the water contact angle under the above-mentioned conditions, we observed that the water contact angle decreased by 30° or more compared to that of the neat HPEB90C10, except for the 2.5 wt% PVP concentration that was not well formed in the hydrogel (Figure 6d–f). This indicated that the surface of the hydrophobic polyurethane was well-changed into a biocompatible PVP hydrogel, except at low PVP concentrations. Thus, through the hydrogel amount and swelling ratio experiments, the optimum hexane treatment time, PVP concentration and EGDMA concentration for hydrogel formation were considered to be 2.5 min, 5 wt% and 5 wt%, respectively.

The MTT assay was used to measure the cellular metabolic activity of the hydrogel surface-modified PEB90C10 film as an indicator of cell viability. HDFn cells at a density of 5.0 × 10^4^ cells/mL were cultured with submerged substrates (2 × 2 mm) in 96 well plates (Figure 7a). These cells were cultured for 24 h and the cell viability was quantitatively evaluated using MTT assays. The optical density measurements obtained from these assays at 3, 9 and 24 h were converted to relative cell viability values (%) by normalizing to the value at 3 h and the control.

Figure 7b,c are representative images of the HDFn cells and the cells stained by MTT, respectively and indicate that the cells exposed to the hydrogel surface-modified polyurethane copolymer film were still alive. As shown in Figure 7d, the cells grew well and proliferated 1.5-fold more in the cellular metabolic activity at 24 h than that at 3 h. By normalizing with the cellular metabolic activity, the cells exposed to the hydrogel surface-modified HPEB90C10 film showed no significant cytotoxicity up to 24 h (Figure 7e). Figure 7e shows that following PVP hydrogelation on the polyurethane copolymer film the cell viability remained statistically constant (up to 95%) for 24 h, thereby indicating that the hydrogel surface-modified polyurethane copolymer film displays excellent cell viability.

## 4. Conclusions

In this study, a hydrogel surface-modified polyurethane copolymer film with good mechanical properties, water resistance and biocompatibility was developed. Hydrophobic, water-resistant and mechanically durable polyurethane copolymers were synthesized by copolymerization of 90 mol% HPEB and 10 mol% PC; HPEB and PC played the role in providing the polyurethane copolymer with the water resistance and mechanical durability, respectively, for protecting an inner electronic circuit. A hydrophilic PVP hydrogel layer was formed on the surface of the polyurethane copolymer film through Fenton reaction, resulting in excellent cell viability. This hydrogel surface-modified polyurethane copolymer film allows the long-term stable use of an implantable biomedical device in the human body. Indeed, the hydrophobic inner polyurethane layer in the film protects the electronic circuits inside the package from moisture and electrolytes in the body, while the hydrophilic outer PVP hydrogel layer imparts excellent biocompatibility. This is expected to extend its potential application as an exterior material for flexible and wearable biomedical devices. Currently, studies of a multilayered amphiphilic Janus nanocomposite film to further enhance the water resistance and biocompatibility are underway. 

## Figures and Tables

**Figure 1 micromachines-12-00447-f001:**
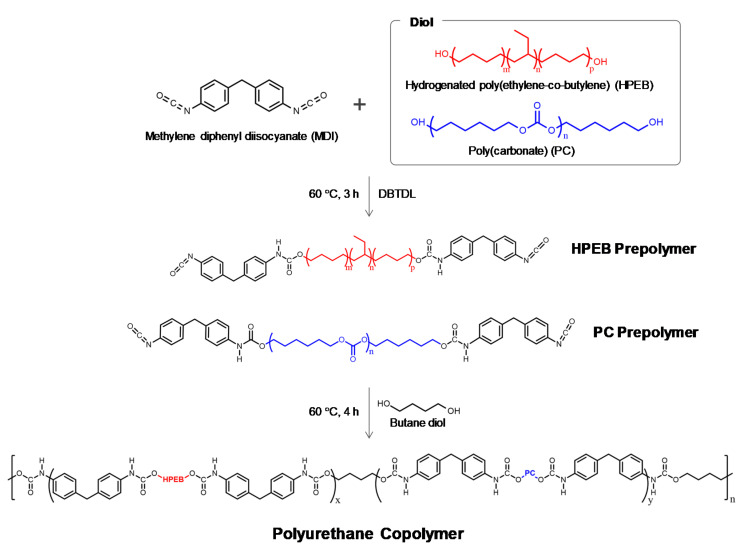
Synthetic scheme for the polyurethane copolymers.

**Figure 2 micromachines-12-00447-f002:**
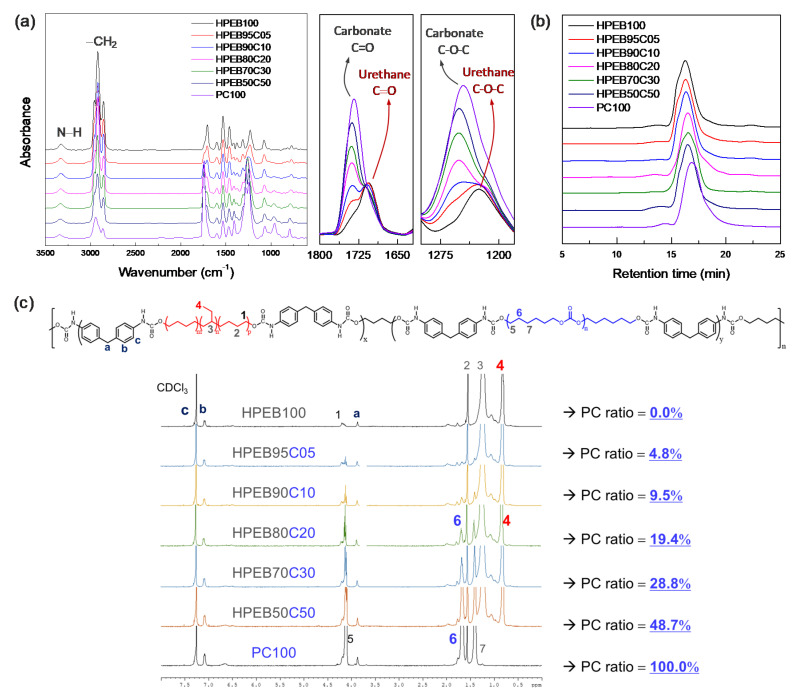
(**a**) FT-IR spectra, (**b**) GPC plots and (**c**) ^1^H-NMR spectra of the polyurethane copolymers.

**Figure 3 micromachines-12-00447-f003:**
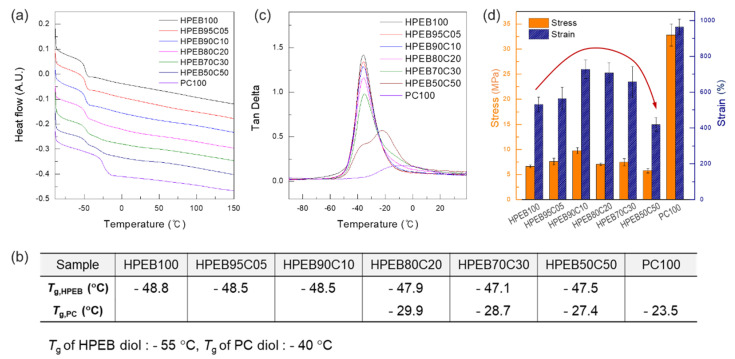
(**a**) DSC thermograms, (**b**) *T*_g_ values measured by DSC, (**c**) Tan δ plots measured by DMA and (**d**) tensile stress and strain values of the polyurethane copolymers.

**Figure 4 micromachines-12-00447-f004:**
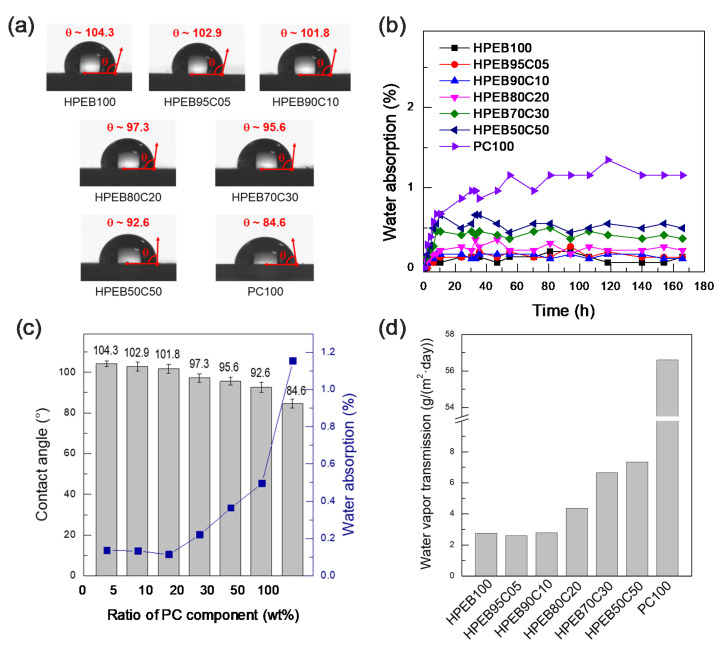
(**a**) Water contact angle images of the polyurethane copolymers, (**b**) water absorption plots of the polyurethane copolymers as a function of time, (**c**) relationship graph between the water contact angle values and the water absorption of the polyurethane copolymers as a function of the amount of PC component and (**d**) water vapor transmission values of the polyurethane copolymers.

**Figure 5 micromachines-12-00447-f005:**
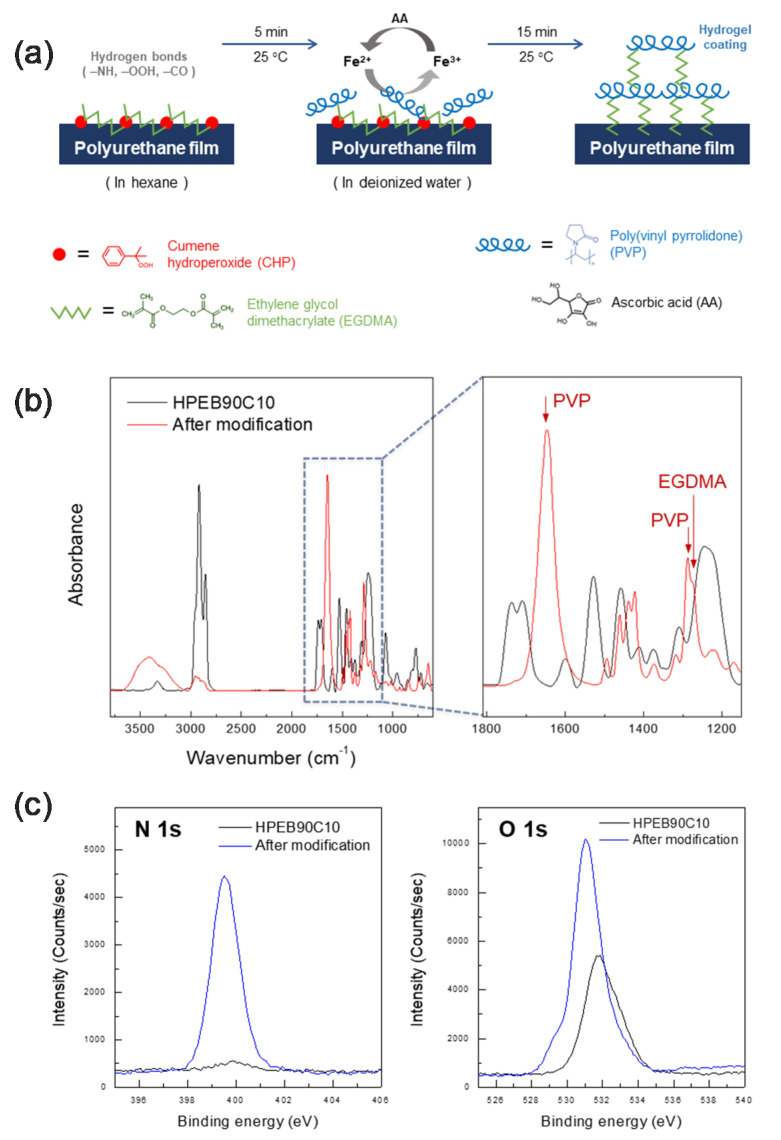
(**a**) Schematic illustration of PVP hydrogelation process using Fenton reaction on the surface of polyurethane copolymers. (**b**) comparison of FT-IR spectra for HPEB90C10 before and after PVP hydrogelation. (**c**) comparison of XPS spectra for HPEB90C10 before and after PVP hydrogelation.

**Figure 6 micromachines-12-00447-f006:**
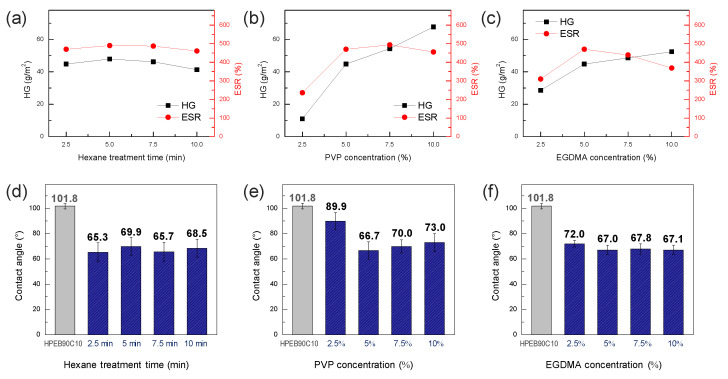
(**a**) Hydrogel amount (HG) values and equilibrium swelling ratio (ESR) values of the PVP hydrogel formed on the HPEB90C10 film as a function of (**a**) the hexane treatment time, (**b**) PVP concentration, (**c**) EGDMA concentration. Water contact angle values of PVP hydrogel-modified HPEB90C10 as a function of (**d**) the hexane treatment time, (**e**) PVP concentration, (**f**) EGDMA concentration.

**Figure 7 micromachines-12-00447-f007:**
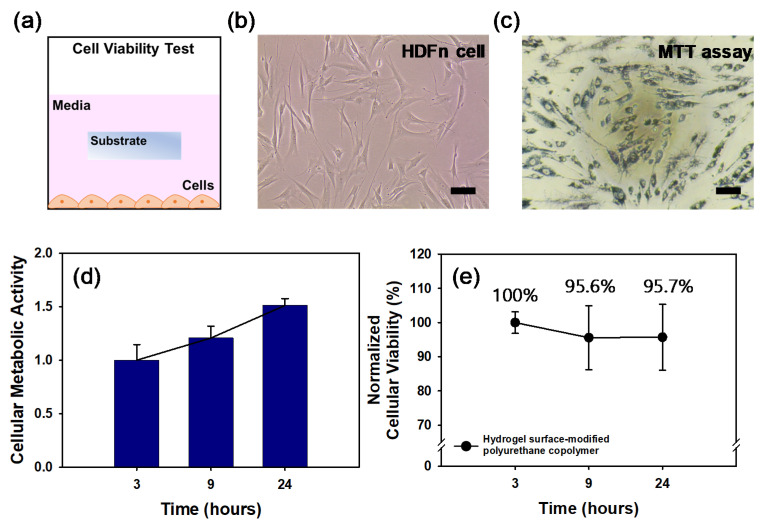
(**a**) Schematic design of cell viability test. Bright field images are Human dermal fibroblasts (HDFn) (**b**) on culture plate and (**c**) MTT assay for cytotoxicity. (scale bar; 100 μm) (**d**) The cellular metabolic activity measured at each time was normalized to the metabolic activity characterized right after cell stabilization on well. (**e**) The cell viability measured at each time was normalized to the characterized pure medium group at each time, comparing to the hydrogel surface-modified polyurethane copolymer.

**Table 1 micromachines-12-00447-t001:** Molar feed ratio of reactants for the polyurethane copolymerization.

Sample	Mol %				Wt% of Hard Segment
	MDI	HPEB	PC	BD	
HPEB100	2.0	1.0	0	1.0	15
HPEB95C05	2.0	0.95	0.05	1.0	15
HPEB90C10	2.0	0.9	0.1	1.0	15
HPEB80C20	2.0	0.8	0.2	1.0	15
HPEB70C30	2.0	0.7	0.3	1.0	15
HPEB50C50	2.0	0.5	0.8	1.0	15
PC100	2.0	0	1.0	1.0	15

**Table 2 micromachines-12-00447-t002:** Average molecular weight and distribution of the polyurethane copolymers.

Sample	*M* _n_	*M* _w_	PDI
HPEB100	64,900	82,200	1.27
HPEB95C05	62,400	80,100	1.28
HPEB90C10	63,000	76,100	1.21
HPEB80C20	60,800	81,700	1.21
HPEB70C30	65,100	79,400	1.22
HPEB50C50	63,400	79,300	1.25
PC100	52,600	69,300	1.32

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
