# Peer review of "Hydrogel Surface-Modified Polyurethane Copolymer Film with Water Permeation Resistance and Biocompatibility for Implantable Biomedical Devices"

_micromachines, 2021, doi:10.3390/mi12040447_

Round 1

Reviewer 1 Report

The manuscript entitled "Hydrogel Surface-Modified Polyurethane Copolymer Film with Water Permeation Resistance and Biocompatibility for Implantable Biomedical Devices" presents the obtaining of a polyurethane copolymer with excellent mechanical properties and high hydrophobicity using hydrogenated poly (ethylene-co-butylene) (HPEB) and polycarbonate (PC). Although the idea is interesting and the experiments in this study are well planned and of good technical quality, several issues need to be addressed for the manuscript to be published in Micromachines. To improve the manuscript I suggest the following comments:

  • the introduction can be improved: a list of hydrophilic polymers used for grafting polyurethane can be added and some references can be added to explain the choice of PVP for this study.
  • the results obtained at FTIR  and H NMR in the case of polymerization of polyurethane should contain some references.
  • Authors should also include references to the obtained results to strengthen their discussions

Author Response

Attached is the response to reviewer 1.

Reviewer 2 Report

The manuscript, “Hydrogel Surface-Modified Polyurethane Copolymer Film with Water Permeation Resistance and Biocompatibility for Implantable Biomedical Devices” by Hey In Jeong et al, describes the polymeric material synthesis and characterization aiming for implantable devices packaging. The material system is based on polyurethane copolymer with external PVP hydrogel coating. The authors have performed a full characterization of the polymer as well as in vitro analysis. The manuscript should get published eventually after addressing my following comments:

  1. Implantable packaging has two approaches in general: hermetic seal by using metal/glass/ceramic and non-hermetic seal by using polymeric materials. There are pros and cons for both methods. It is recommended to add the following two references for each method together with ref 8 and ref 9. Meanwhile, will the material discussed in this manuscript aim for long-term or short-term implantable devices?

1) Yin, M., Borton, D.A., Aceros, J., Patterson, W.R. and Nurmikko, A.V., 2013. A 100-channel hermetically sealed implantable device for chronic wireless neurosensing applications. IEEE transactions on biomedical circuits and systems, 7(2), pp.115-128

2) Wang, P., Lachhman, S.B., Sun, D., Majerus, S.J.A., Damaser, M.S., Zorman, C.A., Feng, P.L. and Ko, W.H., 2013, January. Non-hermetic micropackage for chronic implantable systems. In International Symposium on Microelectronics (Vol. 2013, No. 1, pp. 000166-000170). International Microelectronics Assembly and Packaging Society.

  1. Formatting:

The references had some format issues. Please modify.

All the table look low resolution. Please improve the resolution.

Line 156 “4 mL” is not a correct unit.

  1. In the manuscript, it is not clear what’s the advantage of creating polyurethane-PC polymers and to what extent compared with polyurethane. It is recommended that a more detailed summary of the improvement of the copolymer can be stated in the summary part.
  2. In terms of the cell test, what’s the result of putting copolymer in the cell solution without the gel? In terms of the experiment time length, how 24-hour is decided and what about longer time performance, say a week?
  3. Lin 116-118, in terms of copolymer manufacturing process, the author group used melt-pressing process. How to apply the polymer as the coating material for real implantable electronic devices and sensors? Is the temperature and pressure too high for real implantable device packaging process?
  4. In terms of the WVTR results, what’s the requirement for the packaging material for water permeation and can we say 3 g/m2-day satisfy the implant packaging requirement for short or long term packaging applications?

Author Response

Attached is the response to reviewer 2
